# Polyketide Derivatives, Guhypoxylonols A–D from a Mangrove Endophytic Fungus *Aspergillus* sp. GXNU-Y45 That Inhibit Nitric Oxide Production

**DOI:** 10.3390/md20010005

**Published:** 2021-12-21

**Authors:** Xiaoya Qin, Jiguo Huang, Dexiong Zhou, Wenxiu Zhang, Yanjun Zhang, Jun Li, Ruiyun Yang, Xishan Huang

**Affiliations:** 1State Key Laboratory for Chemistry and Molecular Engineering of Medicinal Resources, Collaborative Innovation Center for Guangxi Ethnic Medicine, College of Chemistry and Pharmaceutical Sciences, Guangxi Normal University, Guilin 541005, China; qinxiaoya6536@163.com (X.Q.); zhoudexiong3@163.com (D.Z.); wenxiuz912@163.com (W.Z.); lijun9593@gxnu.edu.cn (J.L.); 2School of Chemical Engineering and Technology, Guangdong Industry Polytechnic, Guangzhou 510300, China; huangjiguo@126.com; 3Guangxi Key Laboratory of Green Chemical Materials and Safety Technology, Beibu Gulf University, Qinzhou 535011, China; Zhangyj201608@163.com

**Keywords:** *Aspergillus* sp., mangrove endophytic fungus, guhypoxylonols A–D, anti-inflammatory

## Abstract

Four undescribed compounds, guhypoxylonols A (**1**), B (**2**), C (**3**), and D (**4**), were isolated from the mangrove endophytic fungus *Aspergillus* sp. GXNU-Y45, together with seven previously reported metabolites. The structures of **1**–**4** were elucidated based on analysis of HRESIMS and NMR spectroscopic data. The absolute configurations of the stereogenic carbons in **1**–**3** were established through a combination of spectroscopic data and electronic circular dichroism (ECD). Compounds **1**–**11** were evaluated for their anti-inflammatory activity. Compounds **1**, **3**, **4**, and **6** showed an inhibitory activity against the production of nitric oxide (NO), with the IC_50_ values of 14.42 ± 0.11, 18.03 ± 0.14, 16.66 ± 0.21, and 21.05 ± 0.13 μM, respectively.

## 1. Introduction

Marine-derived endophytic fungi have drawn considerable attention for drug discovery, and have been shown to produce various constituents, including sesquiterpenes, alkaloids, and polyketides [1]. Fungi are prolific producers of a variety of biologically active secondary metabolites, including anti-inflammatory, antibiotics, and cytotoxic compounds [1,2]. Lately, the investigation of the constituents of a fungus *Pleosporales* sp., isolated from diverse marine environments has led to the discovery of broad-spectrum cytotoxic secondary metabolites, such as dipleosporalones A and B [3]. In recent years, metabolites discovered from marine-derived fungi have been shown to display a broad range of promising biological activities [1,2,3,4,5,6]. Our group has reported a series of polyketides and structurally related polyketide derivatives from the culture of mangrove endophytic fungi [7,8,9,10].

As part of our ongoing project to discover anti-inflammatory polyketide derivatives from mangrove endophytic fungi, modifications of the composition of the culture medium were employed to reinvestigate the secondary metabolites of *Aspergillus* sp. GXNU-Y45, isolated from a fresh branch of the mangrove plant *Acanthus ilicifolius L*. Chemical investigation of its culture extracts resulted in the isolation of four undescribed polyketides, guhypoxylonols A (**1**), B (**2**), C (**3**), and D (**4**), together with seven previously reported metabolites (**5**–**11**) (Figure 1). Preliminarily screening of **1**–**11** in Appendix A for their ability to prevent NO production of lipopolysaccharide (LPS)-stimulated RAW264.7 cells showed that **1**, **3**, **4**, and **6** have significant inhibitory potency. Herein we report the details of isolation, structure elucidation, and anti-inflammatory activity evaluation of **1**, **3**, **4**, and **6**.

## 2. Results and Discussion

### 2.1. Structure Elucidation of the Compounds

Compound (**1**) was obtained as a brown oil. The molecular formula C_2__1_H_1__8_O_6_ was determined from the quasimolecular ion at *m*/*z* 389.1004 ([M + Na]^+^, calcd for C_2__1_H_1__8_O_6_Na, 389.1001) from a high resolution electrospray ionization mass spectrum (HRESIMS) and the ^13^C NMR spectrum (Table 1). The ^1^H NMR spectrum of **1** displayed two multiplets at *δ*_H_ 2.50 (1H, H-2α), and 1.68 (1H, H-2β), one multiplet at *δ*_H_ 5.22 (1H, H-1), one triplet at *δ*_H_ 4.74 (1H, H-3), two double doublets at *δ*_H_ 3.94 (1H, H-6b), and *δ*_H_ 3.78 (1H, H-7), five aromatic protons at *δ*_H_6.71 (1H, H-5), 7.38 (1H, H-6), 6.84 (1H, H-10), 7.55 (1H, H-11), and 7.43 (1H, H-12), two phenolic hydroxyl protons at *δ*_H_ 9.54 (1H, H-4), and 12.32(1H, H-9). The ^13^C NMR spectrum (Table 1) exhibited 21 carbon signals including one ketone carbonyl at *δ*_C_ 206.5, one methoxyl at *δ*_C_ 55.9, one sp^3^ methylene at *δ*_C_ 39.7, four oxygenated methine sp^3^ at *δ*_C_76.4, 70.4, 62.5, and 56.1, five protonated sp^2^ carbons at *δ*_C_ 136.1, 125.5, 121.5, 115.6, and 112.9, and eight non-protonated sp^2^ carbons at *δ*_C_161.4, 154.4, 134.4, 117.8, 114.0, 138.2, 134.2, 140.0, and 144.9. Analysis of the 2D-NMR spectra (Figure 2) revealed that the structure of **1** resembled that of the previously reported **6** [11] except for the chemical shift value of C-7 which appeared at *δ*_C_ 76.4 CH, indicating that C-7 is oxygen-bearing.

The relative configuration of **1** was determined by the NOESY spectrum (Figure 3) analysis. The NOESY correlations between H-1 (*δ*_H_ 5.22) and OCH_3_-3 (*δ*_H_ 3.29), OCH_3_-3 and H-6b (*δ*_H_ 3.94), and H-6b and OH-7 (*δ*_H_ 6.17) determined the relative configuration of **1** as 1*S**3*S**6b*R**7*S**. The experimental ECD spectrum of **1** was recorded (Figure 4) and the calculated ECD spectrum of 1*S*3*S*6b*R*7*S*-**1** fits well with the experimental ECD spectrum of **1**, as shown in Figure 4. Since **1** has not been previously reported, it was named guhypoxylonol A.

Compound (**2**) was obtained as a colorless powder with a molecular formula of C_1__2_H_1__6_O_3_ as deduced from the HRESIMS *m*/*z* 231.0998 [M + Na]^+^ (cald 231.0997 for C_1__2_H_1__6_O_3_Na), indicating six degrees of unsaturation. The ^1^H-NMR (Table 2) showed two methoxyl singlets at *δ*_H_ 3.31 (3H, s, OCH_3_-4), and 3.75 (3H, s, OCH_3_-5), three aromatic protons at *δ*_H_ 7.24 (1H, d, *J* = 7.9 Hz, H-6), 7.14 (1H, d, *J* = 7.7 Hz, H-8), and 6.83 (1H, d, *J* = 8.1 Hz, H-7), two multiplets at *δ*_H_ 1.80 (2H, m, CH_2_-2), and 1.51, 2.09 (2H, m, CH_2_-3), and two multiplets at *δ*_H_ 4.41 (1H, m, H-1), and 4.35 (1H, m, H-4). The ^13^C NMR spectrum (Table 2) showed 12 carbon signals comprising six aromatic carbons of a benzene ring (*δ*_C_ 157.5 C, 143.1 C, 128.5 CH, 124.8 C, 118.7 CH and 108.8 CH), two methoxyls (*δ*_C_ 55.7 and 56.7), two methylene sp^3^ (*δ*_C_ 27.2 and 24.7), and two oxygenated methine sp^3^ (*δ*_C_ 69.8 and 67.8). The COSY spectrum (Table 2) of **2** displayed two isolated proton spin systems (H-1/H_2_-2/H_2_-3/H-4, and H-6/H-7/H-8). The HMBC spectrum showed correlations from the proton sinal at *δ*_H_ 4.41 (1H, m, H-1) to *δ*_C_ 24.7 (C-3), 118.7 (C-8), and 143.1 (C-8a), from *δ*_H_ 4.35 (1H, t, *J* = 2.8 Hz, H-4) to *δ*_C_ 157.5 (C-5), 27.2 (C-2), and 143.1 (C-8a). The ^1^H and ^13^C NMR spectra of **2** were very similar to those of nodulisporol [12]. The main difference between **2** and nodulisporol was the replacement of a hydroxyl group with a methoxy group at C-4.

The relative configuration of **2** was determined from its NOESY spectrum, which showed correlations from H-1/H-3*α* (*δ*_H_ 2.09), and H-4/H-3*β* (*δ*_H_ 1.51) suggesting that H-1 and H-4 were on the opposite face. To establish the absolute configuration of C-1 and C-4, the ECD spectra of two simplified isomers (1*S*4*S*, and 1*R*4*R*) of **2** were calculated at the Cam-B3LYP/6-31+G(d,p) level of theory in methanol, and these calculated spectra were compared with the experimental spectrum of **2**. The experimental ECD spectrum of **2** showed an excellent fit with the calculated ECD spectrum of 1*S*4*S*-**2** (Figure 4), establishing the absolute configurations of C-1 and C-4 as 1*S*4*S*. Since **2** has never been reported, it was named guhypoxylonol B.

Compound (**3**) was obtained as a colorless powder with a molecular formula of C_13_H_18_O_3_ as deduced from the HRESIMS *m*/*z* 223.1332 [M + H]^+^ (cald 223.1334 for C_13_H_19_O_3_), indicating five degrees of unsaturation. The ^1^H NMR (Table 3), in combination with DEPT and HSQC spectra, displayed two doublets of methylene group at *δ*_H_ 4.65 (*J* = 15.8 Hz, H-8) and 4.58 (*J* = 15.8 Hz, H-8), two multiplets of methine groups at *δ*_H_ 3.86 (*J* = 6.6, 2.6 Hz, H-2) and 2.63 (*J* = 6.8, 2.6 Hz, H-3), two methyl doublets at *δ*_H_ 1.18 (*J* = 6.8 Hz, H-11) and *δ*_H_ 1.19 (*J* = 6.6 Hz, H-12), and two methyl singlets at *δ*_H_ 2.10 (H-9, H-10). The ^13^C NMR (Table 3) spectrum, in combination with HMQC spectrum, of **3** revealed the presence of four methyl carbons at *δ*_C_ 21.0, 18.2, 9.1, and 11.1, one sp^3^ methylene carbon at *δ*_C_ 60.8, two sp^3^ methine carbons at *δ*_C_ 76.0 and 36.4, together with six non-protonated sp^2^ carbons at *δ*_C_ 153.3, 149.6, 134.8, 115.9, 114.4, and 111.3. The COSY (Figure 2) correlations from H-2 to H-3 and H_3_-11, and H-3 to H_3_-12 suggest the existence of -CH(CH_3_)CH(CH_3_)O-. The HMBC (Figure 2) correlations from H-2 to *δ*_C_ 21.0 (C-11), 134.8 (C-3a), 36.4 (C-3), and 60.8 (C-8), from H-3 to *δ*_C_ 134.8 (C-3a), 115.9 (C-4), 114.4 (C-7a), 18.2 (C-12), and 21.0 (C-11), suggests that C-3 is connected to C-3a. The HMBC correlations from H-9 (*δ*_H_ 2.10) to C-4, C-5 (*δ*_C_ 111.3), and C-3a, from H-10 (*δ*_H_ 2.10) to C-4, C-5, C-6 (*δ*_C_ 153.3), indicate that the two methyl groups were on C-4 and C-5, respectively. Finally, the HMBC correlations from H-8 to C-3a, C-7a, C-2 (*δ*_C_ 76.0), and C-7 (*δ*_C_ 149.6), indicated that the remaining substructure of **3** was established as shown in Figure 1.

A NOSEY correlation observed between H-2 and H-3, suggests that the relative configuration of **3** is either 2*R**3*R** or 2*S**3*S** (Figure 3). The absolute configurations of C-2 and C-3 were established by comparing the experimental and calculated ECD spectra of 2*R*3*R*, and 2*S*3*S*. The experimental ECD spectrum of **3** matched very well with the calculated 2*S*3*S*-**3** ECD spectrum (Figure 4), calculated at the Cam-B3LYP/6-311+G (2d, p) level of theory in methanol. Therefore, the absolute configurations of C-2 and C-3 were determined to be 2*S*3*S*. Since **3** has never been reported, it was named guhypoxylonol C.

Compound (**4**) was obtained as a white powder and the molecular formula C_25_H_30_O_9_ was deduced from the HRESIMS *m*/*z* 473.1816 [M − H]^−^ (cald 473.1812 for C_25_H_29_O_9_), indicating 11 degrees of unsaturation. The ^1^H NMR (Table 4) spectrum of **4** displayed two methyl singlets at *δ*_H_ 2.10 (H-9) and 2.07 (H-10), one methoxyl singlet at *δ*_H_ 3.67 (-OCH_3_-8), and two singlets at *δ*_H_ 3.73 (H_2_-7) and 2.50 (H_2_-11). The ^13^C NMR spectrum (Table 4), in combination with the HSQC spectrum of **4**, displayed one ketone carbonyl at *δ*_C_ 207.9 (C-12), one ester carbonyl at *δ*_C_ 173.8 (C-8), one methoxy at *δ*_C_ 52.5 (OCH_3_), two methyls at *δ*_C_12.1, and 9.0, and the two sp^3^ methylene carbons at *δ*_C_ 36.5 (C-7) and 32.5 (C-11). The presence of six non-protonated sp^2^ at *δ*_C_123.1, 118.7, 155.3, 112.5, 157.5, and 130.6 is an indicative of the presence of a benzene ring. The HMBC correlations (Figure 2) from *δ*_H_ 3.73 (H-7) to C-8, 123.1 (C-6), 118.7 (C-1), and from *δ*_H_ 3.67 to C-8, confirm that a methyl acetate is connected to C-1. HMBC correlations from *δ*_H_ 2.07 (H-9) to C-1, 130.6 (C-2), and 157.5 (C-3), from *δ*_H_ 2.07 (H-10) to *δ*_C_112.5 (C-4), 155.3 (C-5), and C-3, and from H-11 to C-1 and C-12, suggested that **4** contains methyl (3,5-dihydroxy-2,4-dimethyl pheny) acetate moiety, with -CH_2_-C=O connected to C-6. Since the molecular formula of C_25_H_30_O_9_, only a ketone carbonyl (*δ*_C_207.9) is present in **4**. Therefore, the structure of **4** is a disubstituted acetone whose substituents are methyl (3,5-digydroxy-2,4-dimethylphenyl)acetate. Since **4** has never been reported, it was named guhypoxylonol D.

The previously described **5**–**11** were identified based on the analysis of their NMR data, and compared with those reported in the literature and identified as hypoxylonol C (**5**) [11], hypoxylonol B (**6**) [11], daldinone C (**7**) [13], nodulisporol (**8**) [12], isosclerone (**9**) [14], xylarenone (**10**) [14], scytalone (**11**) [15], respectively.

### 2.2. Anti-Inflammatory Activity

Compounds **1**–**11** were evaluated for their anti-inflammatory effects on the production of the NO in the RAW 264.7 macrophage cell line exposed to the inflammatory stimulus by lipopolysaccharide (LPS) (Table 5). Compounds **1**, **3**, **4**, and **6** showed inhibitory activity against the production of NO, with the IC_50_ values 14.42 ± 0.11, 18.03 ± 0.14, 16.66 ± 0.21, and 21.05 ± 0.13 μM, respectively. Dexamethasone was used as a positive control with IC_50_ value of 16.12 ± 1.41 μM, while **2**, **5**, and **7**–**11** did not show any inhibitory activity under their safe concentrations. 

## 3. Materials and Methods

### 3.1. General Experimental Procedures

NMR spectra were recorded on a AVANCE-400 spectrometer (Bruker, Bremen, Germany). The chemical shifts of ^1^H and ^13^C NMR spectra are given in *δ* (ppm) and referenced to the solvent signal (DMSO-*d*_6_, *δ*_H_ 2.50 and *δ*_C_ 39.52, CD_3_OD-*d*_4_, *δ*_H_ 3.34 and *δ*_C_ 49.00). Coupling constants (*J*) are reported in Hz. The mass spectrometric (HRESIMS) data were acquired using a Micro Mass Q-TOF spectrometer (Waters Corporation, Milford, MA, USA). ECD data was recorded using a JASCO J-715 spectropolarimeter (Jasco, Tokyo, Japan). Semipreparative HPLC was performed on an ODS column (10 × 250 mm, 5 µm, 3 mL/min, YMC, Kyoto, Japan).

### 3.2. Fungal Material

The strain GXNU-Y45 was isolated from a leaf of a mangrove tree *Acanthus ilicifolius*, October 2019, in Beihai City, China. The fungal strain GXNU-Y45 was identified as *Aspergillus* sp. based on the sequence of its internal transcribed spacer region (ITS) and morphology. ITS-rDNA of GXNU-Y45 was submitted to GenBank and the accession number is MT626059.

### 3.3. Fermentation, Extraction, and Isolation

The fungus was cultured in 60 × 1000 mL Erlenmeyer flasks each containing 50 g cooked rice and 60 mL of water (30 g sea salt, per liter pure water) or 300 mL medium (liquid media, 20.0 g dextrose, 20.0 g potatoes, 30 g sea salt, per liter pure water). The fungus was cultured in the medium and incubated at room temperature for 35 days.

### 3.4. Extraction and Isolation

The fermented material was extracted three times with EtOAc to obtain 16.8 g crude extract (liquid medium) and 20.2 g (solid medium). The crude extract was subjected to a silica gel VLC column, eluting with a stepwise gradient of petroleum ether-EtOAc (10:1, 8:1, 6:1, 4:1, 2:1, 1:1, *v*/*v*) to yield six subfractions (Fr. 1–Fr. 6). Fr. 3 (3 g) was applied to ODS silica gel with gradient elution of MeOH-H_2_O (3:7, 4:6, 5:5, 6:4, 7:3, 9:1, 0:1, *v*/*v*) to afford four subfractions (Fr. 3-1–Fr. 3-4). Fr. 3-2 (650 mg) was subjected to semipreparative HPLC (70% MeOH/H_2_O; 3 mL/min) to obtain **1** (15.6 mg), **2** (7.5 mg), and **3** (4.4 mg). Fr. 3-3 (345 mg) was repurified by RP-18 CC (eluted with MeOH/H_2_O from 3:7 to 10:0, *v*/*v*) and Sephadex LH-20 (eluted with CH_2_Cl_2_/MeOH, 5:5, *v*/*v*) to afford **5** (10.6 mg), **9** (3.3 mg), **10** (5.2 mg), and **11** (6.7 mg). Fr. 4 (1.1 g) was separated by ODS silica gel with gradient elution of MeOH-H_2_O (1:9, 2:8, 3:7, 4:6, 5:5, 6:4, 7:3, 9:1, 0:1, *v*/*v*) to yield four subfractions (Fr. 4-1–Fr. 4-4). Fr.4-3 (73 mg) was purified by Sephadex LH-20 eluted with CH_2_Cl_2_/MeOH (50:50) to give **4** (6.3 mg). Fr.4-4 (84 mg) was separated by semipreparative HPLC (80% MeCN/H_2_O; 3 mL/min) to give **6** (5.6 mg), **7** (8.1 mg), and **8** (5.2 mg).

Guhypoxylonol A (**1**): was obtained as a brown oil; ${\left[\alpha \right]}_{D}^{20}$ + 63.2 (c0.6, MeOH); ^1^H and ^13^C NMR data (see Table 1 and Table 2); HRESIMS *m*/*z* 389.1004 ([M + Na]^+^ (cald C_2__1_H_1__8_O_6_Na, 389.1001). 

Guhypoxylonol B (**2**): was obtained as a colorless powder; ${\left[\alpha \right]}_{D}^{20}$ + 8.5 (c0.6, MeOH); ^1^H and ^13^C NMR data (see Table 1 and Table 2); HRESIMS *m*/*z* 231.0998 [M + Na]^+^ (cald 231.0997 for C_1__2_H_1__6_O_3_Na).

Guhypoxylonol C (**3**): white powder; ${\left[\alpha \right]}_{D}^{20}$ + 80 (c0.6, MeOH); ^1^H and ^13^C NMR data (see Table 1 and Table 2); HRESIMS *m*/*z* 223.1332 [M + H]^+^ (cald 223.1334 for C_13_H_19_O_3_).

Guhypoxylonol D (**4**): white powder; ^1^H and ^13^C NMR data (see Table 1 and Table 2); HRESIMS *m*/*z* 473.1816 [M − H]^−^ (cald 473.1812 for C_25_H_30_O_9_).

### 3.5. Anti-Inflammatory Assay

The anti-inflammatory effects of compounds **1**–**11** were examined on the production of the NO in LPS-stimulated cells using a method described in the literature [16].

## 4. Conclusions

The chemical investigation of a marine-derived fungus *Aspergillus* sp. GXNU-Y45 resulted in the isolation of four undescribed compounds (**1**–**4**), and seven previously reported metabolites (**5**–**11**). Based on modifications of the culture medium strategy, the fungus *Aspergillus* sp. GXNU-Y45 was cultured in different media to stimulate a production of its metabolites. It was found that the fungus *Aspergillus* sp. GXNU-Y45 produced different metabolites in two culture media. The liquid medium can stimulate the fungus to produce a series of metabolites, **1**, **5**, **6**, **7**, **8**, **9**, **10**, and **2** (a new precursor of **1**). On the contrary the solid medium yeiled **3** and **4**. Different compositions of the culture media represented a powerful tool to induce new metabolites from microorganisms. Preliminarily screening of **1**–**11** for their ability to prevent NO production of LPS-induced RAW264.7 cells showed that **1**, **3**, **4**, and **6** exhibited significant inhibitory effects against NO release with IC_50_ values of 14.42 ± 0.11, 18.03 ± 0.14, 16.66 ± 0.21, and 21.05 ± 0.13 μM, respectively. The inhibition of NO production by **1** and **6** was stronger than **5** and **7**, which showed the same skeleton but differ only the presence of -OCH_3_ at C-3. Compounds **2** and **8**–**11**, which are precursors of **1**, **5**, **6**, and **7**, did not exhibit inhibitory effects against NO release. Compounds **3** and **4** exhibited remarkable inhibitory effects against NO release suggesting that the fully substituted benzene ring was essential for inhibition of the production of NO release. In summary, this study revealed that **1**, **3**, **4**, and **6** could be considered as potential metabolites for further anti-inflammatory studies. 

## Figures and Tables

**Figure 1 marinedrugs-20-00005-f001:**
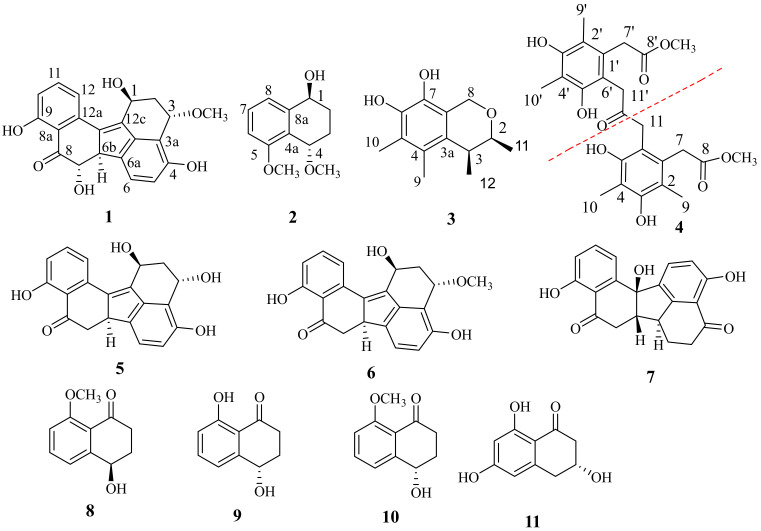
Structures of **1**–**11**.

**Figure 2 marinedrugs-20-00005-f002:**
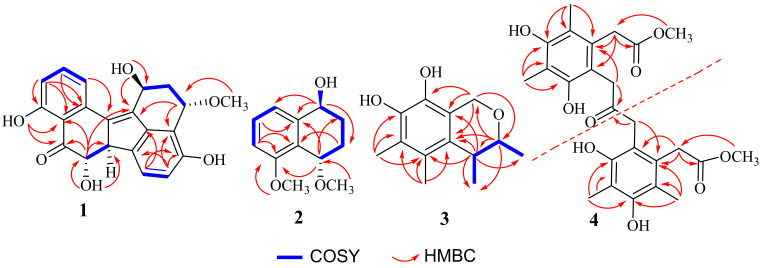
Key COSY of **1-3** and HMBC correlations of **1**–**4**.

**Figure 3 marinedrugs-20-00005-f003:**
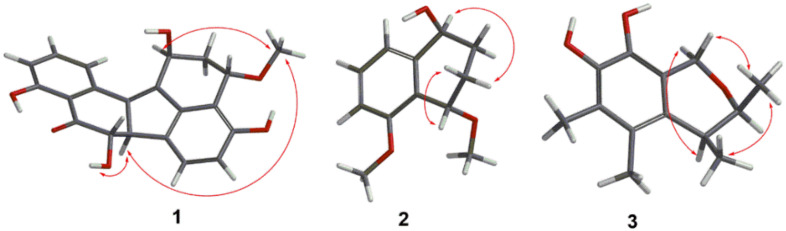
Key NOESY correlations in **1**–**3**.

**Figure 4 marinedrugs-20-00005-f004:**
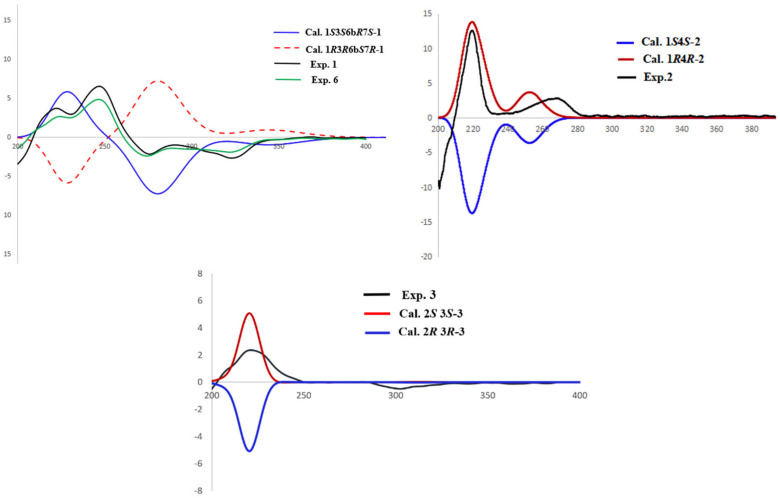
Experimental ECD and calculated ECD spectra of **1**–**3**.

**Table 1 marinedrugs-20-00005-t001:** ^1^H and ^13^C NMR (DMSO-*d_6_*, 600 and 150 MHz) and COSY and HMBC assignment of **1**.

Position	*δ*_C_, Type	*δ*_H_, (Mult., *J* in Hz)	COSY	HMBC
1	62.5, CH	5.22, m	H-2	
2α 2β	39.7, CH_2_	2.50, m 1.68, m	H-1, 3	
3	70.4, CH	4.74, t (3.0)	H-2	C-3a, 12c
4	154.4, C			
5	112.9, CH	6.71, d (8.0)	H-6	C-3a, 4, 6a
6	125.5, CH	7.38, d (8.0)	H-5	C-4, 12d
6a	134.4, C			
6b	56.1, CH	3.94, dd (12.4, 3.1)	H-7	
7	76.4, CH	3.78, dd (12.3, 5.6)	H-6b	C-6b, 8, 8a, 12c
8	206.5, C			
8a	114.0, C			
9	161.4, C			
10	115.6, CH	6.84, d (8.2)	H-11	C-8a, 9, 12a
11	136.1, CH	7.55, d (8.0)	H-10, 12	C-12a
12	121.5, CH	7.43, d (7.7)	H-11	C-12b
12a	138.2, C			
12b	134.2, C			
12c	140.0, C			
12d	144.9, C			
1-OH		5.06, d (7.8)		C-1, 12c
4-OH		9.54, s		
7-OH		6.17, d (5.9)		C-6b, 7
9-OH		12.32, s		C-8, 8a
3-OCH_3_	55.9, CH_3_	3.29, s		C-3

**Table 2 marinedrugs-20-00005-t002:** ^1^H and ^13^C NMR (DMSO-*d_6_*, 600 and 150 MHz) and COSY and HMBC assignment of **2**.

Position	*δ*_C_, Type	*δ*_H_ (Mult., *J* in Hz)	COSY	HMBC
1	67.8, CH	4.41, m	H-2	C-3, 8, 8a
2	27.2, CH_2_	1.80, m	H-1, 3	
3α 3β	24.7, CH_2_	2.09, m 1.51, m	H-2, 4	C-4a
4	69.8, CH	4.35, t (2.8)	H-3	C-2, 5, 8a
4a	124.8, C			
5	157.5, C			
6	128.5, CH	7.24, d (7.9)	H-7	C-4a, 5
7	108.8, CH	6.83, d (8.1)	H-6, 8	C-8a
8	118.7, CH	7.14, d (7.7)	H-7	
8a	143.1, C			
1-OH		5.28, s		
4-OCH_3_	56.7, CH_3_	3.31, s		C-4
5-OCH_3_	55.7, CH_3_	3.75, s		C-5

**Table 3 marinedrugs-20-00005-t003:** ^1^H and ^13^C NMR (CD_3_OD, 400 and 100 MHz) and COSY and HMBC assignment of **3**.

Position	*δ*_C_, Type	*δ*_H_ (Mult., *J* in Hz)	COSY	HMBC
2	76.0, CH	3.86, qd (6.6, 2.6)	H-3, 11	C-3, 3a, 8, 12
3	36.4, CH	2.63, qd (6.8, 2.6)	H-2, 12	C-3a, 4, 7a, 11, 12
3a	134,8, C			
4	115.9, C			
5	111.3, C			
6	153.3, C			
7	149.6, C			
7a	114.4, C			
8	60.8, CH_2_	4.65, d (15.2) 4.58, d (15.2)		C-2, 3a, 7, 7a
9	11.1, CH_3_	2.10, s		C-3a, 4, 5
10	9.1, CH_3_	2.10, s		C-4, 5, 6
11	21.0, CH_3_	1.18, d (6.8)		
12	18.2, CH_3_	1.19, d (6.6)		

**Table 4 marinedrugs-20-00005-t004:** ^1^H and ^13^C NMR (CD_3_OD, 400 and 100 MHz) and HMBC assignment of **4**.

Position	*δ*_C_, Type	*δ*_H_ (Mult., *J* in Hz)	HMBC
1 (1′)	118.7, C		
2 (2′)	130.6, C		
3 (3′)	157.5, C		
4 (4′)	112.5, C		
5 (5′)	155.3, C		
6 (6′)	123.6, C		
7 (7′)	36.5, CH_2_	3.73, s	C-1 (1′), 6 (6′), 8 (8′)
8 (8′)	173.8, C		
9 (9′)	9.0, CH_3_	2.10, s	C-1 (1′), 2 (2′), 3 (3′)
10 (10′)	12.1, CH_3_	2.07, s	C-3 (3′), 4 (4′), 5 (5′)
11 (11′)	32.5, CH_2_	2.50, s	C-1 (1′), 12
12	207.9, C		
8-OCH_3_	52.5, CH_3_	3.67, s	C-8 (8′)

**Table 5 marinedrugs-20-00005-t005:** Inhibitory activities of **1**–**11** on NO production in LPS-induced RAW 264.7 cells ^a^.

Compounds	IC_50_ (μM)
**1**	14.42 ± 0.11
**2**	32.48 ± 0.19
**3**	18.03 ± 0.14
**4**	16.66 ± 0.21
**5**	>80
**6**	21.05 ± 0.13
**7**	>80
**8**	>80
**9**	>80
**10**	>80
**11**	>80
**Dexamethasone** ^b^	16.12 ± 1.41 μM

^a^ Values present mean ± SD of triplicate experiments. ^b^ Dexamethasone was used as a positive control.

## Data Availability

The authors declare that all data of this study are available within the article and its Appendix A file or from the corresponding authors upon request.

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
