# Peer review of "Polyketide Derivatives, Guhypoxylonols A–D from a Mangrove Endophytic Fungus Aspergillus sp. GXNU-Y45 That Inhibit Nitric Oxide Production"

_marinedrugs, 2021, doi:10.3390/md20010005_

Round 1

Reviewer 1 Report

The paper of X.Qion et al describes the isolation, structure elucidation and ani-inflammatory activity of 4 new secondary metabolites from Aspergillus sp.

The “isolation and structure elucidation” part is done very well by using state- of- the-art techniques (1D and 2D-NMR and HR-MS) and by a careful analysis of the experimental data thereby obtained.

The manuscript reports a lot of structural details and NMR-data in Supplementary material seem to support all their structural assignments

I wonder whether the detailed and long description of the NMR spectra of compound 1 is really necessary by considering that it is just a small structural variant of the already reported compounds 5 and 6.

The authors are also invited to propose an explanation why the theoretical ECDs of compounds  1and are in good agreement with their experimental CD spectra (in terms of matching of the wavelengths at which the maximum Cotton effects were observed) whilst this agreement is quite poor with compound 3; indeed, the theoretical maximum wavelength was calculated to be at about 220 nm whilst the experimental value is significantly higher

Author Response

Dear Professor Emilia Ruan:

Thank you very much for your reply. Herein, we submitted the paper (Title: New Anti-inflammatory Polyketide Derivatives, Guhypoxylonols A-D from Mangrove Endophytic Fungus Aspergillus sp. GXNU-Y45, Manuscript ID: marinedrugs-1494557), has been revised as in the attachment. And response the comments as following:

Reviewer1

The paper of X.Qion et al describes the isolation, structure elucidation and ani-inflammatory activity of 4 new secondary metabolites from Aspergillus sp.

The “isolation and structure elucidation” part is done very well by using state- of- the-art techniques (1D and 2D-NMR and HR-MS) and by a careful analysis of the experimental data thereby obtained.

The manuscript reports a lot of structural details and NMR-data in Supplementary material seem to support all their structural assignments

I wonder whether the detailed and long description of the NMR spectra of compound 1 is really necessary by considering that it is just a small structural variant of the already reported compounds 5 and 6.

Response: Thank you for your suggestion. The description of the NMR spectra of compound 1 has been revised and highlighted in the text.

The authors are also invited to propose an explanation why the theoretical ECDs of compounds  1 and are in good agreement with their experimental CD spectra (in terms of matching of the wavelengths at which the maximum Cotton effects were observed) whilst this agreement is quite poor with compound 3; indeed, the theoretical maximum wavelength was calculated to be at about 220 nm whilst the experimental value is significantly higher

Response: Thank you for your suggestions. The theoretical ECDs of compound 3 has been calculated again at the Cam-B3LYP/6-31+G (d, p) level of theory in methanol and the result of the calculation was as figure 4.

Best wishes

Xishan Huang

2021.12.06

Reviewer 2 Report

  1. It would be interesting to see the cytotoxic concentrations of new compounds on macrophages RAW 264 and  compare them with inhibitory activities on NO.
  2. I would suggest replacing the “Anti-inflammatory Polyketide derivatives…”  in the title of this article with “New Polyketide derivatives inhibiting nitric oxide (NO) production…”. For example, article 16 from the list of references cited by the authors is called "... saponins with anti-inflammatory activity ..." and this article studies various indicators of inflammation: the secretion of prostagalandin E2, IL-6, THF-a, the release of nitric oxide by various other methods.
  3. The list of references is incorrectly formatted. I ask the authors to check it out.

Author Response

Dear Professor Emilia Ruan:

Thank you very much for your reply. Herein, we submitted the paper (Title: New Anti-inflammatory Polyketide Derivatives, Guhypoxylonols A-D from Mangrove Endophytic Fungus Aspergillus sp. GXNU-Y45, Manuscript ID: marinedrugs-1494557), has been revised as in the attachment. And response the comments as following:

Reviewer2

  1. It would be interesting to see the cytotoxic concentrations of new compounds on macrophages RAW 264 and compare them with inhibitory activities on NO.

Response: Thank you for your suggestions. A mistake has been corrected in the table3,the initial cytotoxic concentration was 80μM.

  1. I would suggest replacing the “Anti-inflammatory Polyketide derivatives…”  in the title of this article with “New Polyketide derivatives inhibiting nitric oxide (NO) production…”. For example, article 16 from the list of references cited by the authors is called "... saponins with anti-inflammatory activity ..." and this article studies various indicators of inflammation: the secretion of prostagalandin E2, IL-6, THF-a, the release of nitric oxide by various other methods.

Response:  Thank you for your suggestions. The title was revised and highlighted in the text.

  1. The list of references is incorrectly formatted. I ask the authors to check it out.

Response: Thank you for your suggestions. The list of references was revised and highlighted in the text.

Best wishes

Xishan Huang

2021.12.06

Round 2

Reviewer 2 Report

The method for measuring nitric oxide is not very conveniently described. For the future, I would recommend at least briefly describing the methodology.

Author Response

Dear Professor Emilia Ruan:

Thank you very much for your reply. Herein, we submitted the paper (Title: New Anti-inflammatory Polyketide Derivatives, Guhypoxylonols A-D from Mangrove Endophytic Fungus Aspergillus sp. GXNU-Y45, Manuscript ID: marinedrugs-1494557), has been revised as in the attachment. And response the comments as following:

Reviewer2

The method for measuring nitric oxide is not very conveniently described. For the future, I would recommend at least briefly describing the methodology.

Response: thank you for you suggestion. The method for measuring nitric oxide, which was cited in the text, was reported by our research group. Thanks again for your suggestion.

Best wishes

Xishan Huang

2021.12.10